# ANALYSIS AND INTERPRETATION OF DEEP CNN REPRESENTATIONS AS PERCEPTUAL QUALITY FEATURES

## ABSTRACT

Pre-trained Deep Convolutional Neural Network (CNN) features have popularly been used as full-reference perceptual quality features for CNN based image quality assessment, super-resolution, image restoration and a variety of image-to-image translation problems. In this paper, to get more insight, we link basic human visual perception to characteristics of learned deep CNN representations as a novel and first attempt to interpret them. We characterize the frequency and orientation tuning of channels in trained image classification deep CNNs (e.g., VGG-16) by applying grating stimuli of different spatial frequencies and orientations as input. We observe that the behavior of CNN channels as spatial frequency and orientation selective filters can be used to link basic human visual perception models to their characteristics. Doing so, we develop a theory to get more insight into deep CNN representations as perceptual quality features. We conclude that sensitivity to spatial frequencies that have lower contrast masking thresholds in human visual perception and a definite and strong orientation selectivity are important attributes of deep CNN channels that deliver better perceptual quality features.

## 1 INTRODUCTION

Quantifying human perception of image quality has been a subject of significant research for quite some time. Full-reference objective metrics such as the PSNR (Peak Signal to Noise Ratio) and SSIM (Structural Similarity Index) (Wang et al. (2004)), being fair metrics of distortion between two images, are not a satisfactory metrics to measure differences in perceptual quality. Considering the recent interest in the applications of deep CNNs in perception-oriented problems such as super-resolution, image-restoration, frame-interpolation and style-transfer etc, research into effective loss metrics that quantify perceptual quality and help train CNNs in delivering better perceptual quality has become paramount.

The perceptual loss proposed by Johnson et al. (2016) was one of the first to demonstrate how effective the feature representations of pre-trained image classification CNNs could be as features of full-reference perceptual quality, especially when incorporated into loss functions for image restoration. The perceptual loss is now popularly adopted in many image restoration problems such as super-resolution, style transfer, denoising etc. (Ledig et al. (2017),Wang et al. (2018),Gatys et al. (2016)). Zhang et al. (2018) and Blau & Michaeli (2018) further demonstrate how effective deep CNN representations can be as features of perceptual quality, but without any analysis into their characteristics. More recently, Mechrez et al. (2018) proposed a variation of the perceptual loss called the contextual loss, which still employs deep CNN features as perceptual quality features but uses an approximation of the KL-divergence to quantify distance. The contextual loss has been demonstrated to be quite effective in maintaining natural image statistics during SISR (Single-Image Super-Resolution). The recent PIRM Super-Resolution Challenge Report (Blau et al. (2018)) clearly iterates that the perceptual loss and the contextual loss are the most widely used loss functions for CNN based perceptual image Super-Resolution.

Nevertheless, like most applications of deep learning, there has been little or no effort to understand and interpret the role of deep CNN representations as effective perceptual quality features. This is quite understandable, as it is difficult to find a direction to approach this problem from. Neural networks are non-linear, which makes a tractable analysis tricky. Furthermore, human perception of

quality is also something that is still not understood completely. Most of our basic understanding of human visual perception of quality is in the frequency domain, with models such as the Contrast Sensitivity Function (CSF) (de Faria et al. (1998)). To make a connection between deep CNN features and human perception, it is important to realize that deep CNN channels are essentially complex spatial frequency and orientation selective filters.

We stimulate pre-trained image classification CNNs with sinusoidal grating stimuli, record the response in the form of mean activation of each channel as function of spatial frequency/orientation of input grating, thus quantifying the frequency and orientation selectivity of different channels. This approach makes it significantly easier to establish a connection between perception models such as the CSF with learned deep feature representations. We hypothesize that two attributes are important for deep CNN channels that are good perceptual quality features. The attributes are based on visual masking in human visual perception, which refers to human ability to perceive distortions and contrast in visual stimulus. The first attribute is sensitivity to spatial frequencies at which there is minimal contrast masking in human visual perception (Nadenau et al. (2000)), making the CNN channel sensitive to highly perceivable distortions. The second attribute being a definite and strong orientation selectivity, which helps the channel respond better to image regions with less pattern complexity, where there is less masking for distortions from a perceptual standpoint (Wu et al. (2017)).

We verify our hypothesis by designing an Objective Quality Assessment (OQA) experiment (Sheikh et al. (2006)). OQA experiments correlate the performance of any quality metric with human perception of quality, which is an accepted and standard experimental technique. We group the set of channels in different CNN layers into subsets on the basis of our hypothesis and demonstrate that the group which has channels with our described attributes, delivers a much better as a set of perceptual quality features. We repeat our experiment across multiple layers of many pre-trained image classification networks such as the VGG-16 (Simonyan & Zisserman (2014)), AlexNet (Krizhevsky & Hinton (2012)), ShuffleNet (Zhang et al. (2017)) and SqueezeNet (Iandola et al. (2017)).

## 2  Deep CNN Representations as Perceptual Quality Features

The main motivation behind using pre-trained image classification deep CNN representations as perceptual quality features is that instead of a distance measure between two images being a good FR metric, computing distance after non-linear transformation of images into a high dimensional manifold, might result in a better perceptual quality measure. The high dimensional manifold in this case is the manifold of pre-trained CNN features. The general form for the perceptual loss (Johnson et al. (2016)) is given by Eq. (1)

$$l_p = \frac{1}{M \cdot W \cdot H} \sum_{m=1}^{M} \|\Phi_m^k(I_{\text{out}}) - \Phi_m^k(I_{\text{GT}})\|_2^2 \tag{1}$$

Where '$\Phi_m^k$' is the feature map corresponding to the '$m^{th}$' channel in the '$k^{th}$' layer which as '$M$' number of total channels with feature map dimensions '$H \cdot W$'. As mentioned before, applying pre-trained deep CNN representations as perceptual quality features has proven to be quite effective in FR-IQA methods (Bosse et al. (2017)), image restoration (Wang et al. (2018)) and style transfer (Gatys et al. (2016)) problems, as iterated by Blau et al. (2018).

However, little else is known of the ability and characteristics of deep CNN representations as perceptual quality features. In this work, using basic human perception models, we aim to get more insight into the role of pre-trained deep representations as perceptual quality features.

## 3  Problem Formulation

Section. 2 iterates the motivation and wide spread use of pre-trained deep CNN representations as features of full-reference perceptual quality. However, there has been no effort to explain and interpret the role of deep representations as perceptual features. We consider a CNN convolution layer as collection of channels which deliver perceptual quality features. For example, the *relu3_2* layer of the VGG-16 has 256 channels. Are all of the channels equally effective in delivering good

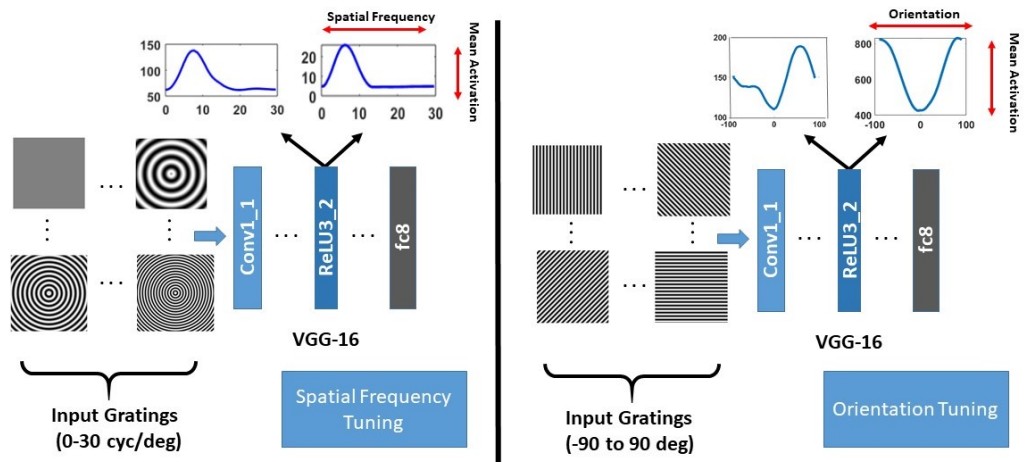

Figure 1: Experimental Setup. The network is stimulated by gratings of varying spatial frequency. The responses of different feature maps are recorded as activation vs spatial frequency data. To quantify orientation tuning, the network is stimulated by gratings of fixed spatial frequency and varying orientations to record mean activation vs orientation data.

perceptual quality features? Are some channels better than others and if so, what attributes make them better?

The problem in question is important in explaining the role of deep CNN representations as perceptual quality features, but it is somewhat difficult to approach because of the 'black-box' nature of neural networks. In section. 4.1, we will introduce a methodology to quantify the spatial frequency and orientation tuning of channels in pre-trained CNNs. Using this formulation, we will interpret and explain deep CNN features as perceptual quality features by making use of basic human visual perception models, which rely on spatial frequency and orientation characteristics of input stimuli. In essence, the formulation in Section. 4.1 will act as a bridge to link attributes of deep representations and basic visual perception.

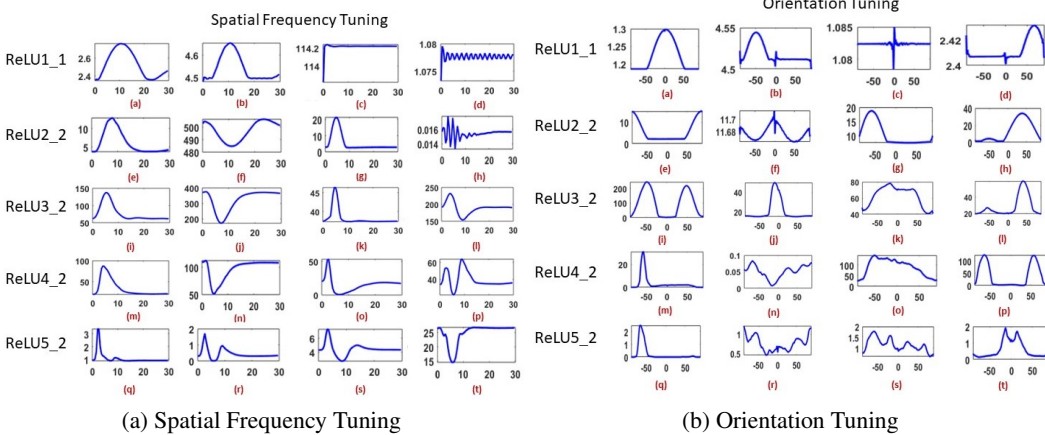

(a) Spatial Frequency Tuning            (b) Orientation Tuning

Figure 2: Characterizing spatial frequency and orientation tuning in channels across different layers of the pre-trained VGG-16.

## 4 A PSYCHOVISUAL APPROACH

### 4.1 FREQUENCY/ORIENTATION TUNING QUANTIFICATION

Our experimental method is inspired by the grating stimulus experiments used by neuro-scientists to study human visual perception characteristics (Kulikowski et al. (1982)). We aim to quantify both the spatial frequency and orientation tuning of different channels in the pre-trained CNN.

To quantify the spatial frequency tuning, we generate concentric sinusoidal gratings of a fixed contrast and varying spatial frequencies (cycles per degree), use them to stimulate pre-trained image classification CNNs and record the responses of the feature maps in the form of mean activation versus spatial frequency for each channel. Fig. 1 illustrates the overall scheme of measuring the spatial frequency responses of channels in various convolution layers of the trained VGG-16 network. The reason we are using a concentric pattern is to eliminate the factor of orientation selectivity from this part of the analysis. Some concentric grating stimulus patterns are shown as input to the trained VGG-16 network in Fig. 2.(a).

To quantify orientation selectivity at low contrast masking thresholds, we stimulate the pre-trained network with linear pattern sinusoidal gratings with different orientations. The gratings have a fixed spatial frequency, which corresponds to the peak of the Contrast Sensitivity Function (CSF). Some sample grating patterns are shown in Fig. 1. Sample observations of orientation selectivity for channels in different layers of the pre-trained VGG-16 are shown in Fig. 2.(b).

### 4.2 VISUAL FREQUENCY SENSITIVITY

In this section, we will use the spatial frequency selectivity quantification in section. 4.1 to introduce the concept of visual frequency sensitivity. Human perception of images is largely dependent on attributes of input stimulus. A significant proportion of neuro-science research advocates the role of the early visual cortex as a spatial frequency analyzer (Maffei & Fiorentini (1973)). Human perception characteristics are dependent on spatial frequency and one of the most significant models that quantifies this characteristic is called the Contrast Sensitivity Function (CSF). The CSF depicts human ability to perceive contrast changes as a function of spatial frequency. The spatial frequencies where the CSF has a higher value, correspond to lower contrast masking thresholds in perception (ease in perceiving distortions and contrast changes). In essence, this corresponds to a higher probability of perceiving distortions at high CSF valued spatial frequencies.

Considering the presented analysis on the spatial frequency selective behavior of deep feature maps. Our hypothesis is that the deep representations that are more sensitive to high CSF valued spatial frequencies, can be better features of perceptual quality. Fig. 3 plots mean activation of two channels versus spatial frequency of the input grating. Feature Map-2 can be seen to have a higher sensitivity compared to Feature Map-1 at high CSF valued spatial frequencies, making Feature Map-2 more sensitive to distortions corresponding to low contrast masking threshold regions in input images.

We model this attribute quantitatively as $\mu_1$ defined in Eq. 2

$$\mu_1(k,m) = \sum_f CSF(f). \left| \frac{\partial a_m^k}{\partial f} \right| \tag{2}$$

where '$k$' is the index for the convolution layer, '$m$' is the feature map index in each convolution layer, '$CSF$' is the contrast sensitivity function (CSF), '$a$' is the mean activation of the feature map and '$f$' is the spatial frequency in cycles per degree. $\mu_1$ quantifies the average sensitivity of a CNN channel weighted by the CSF over different spatial frequencies. The channels having higher $\mu_1$ values should deliver better perceptual features according to our hypothesis, because they can be more sensitive to visually perceivable distortions in input images.

### 4.3 ORIENTATION SELECTIVITY

In addition to the underlying spatial frequency, orientation also plays an important part in human perception of visual stimulus. Neuro-science research indicates that the HVS (Human Visual System) is highly adapted to extract repeated patterns for visual content representation (Wu et al. (2017)). The complexity of a visual pattern has an effect in its perception. If a pattern is regular, the visual

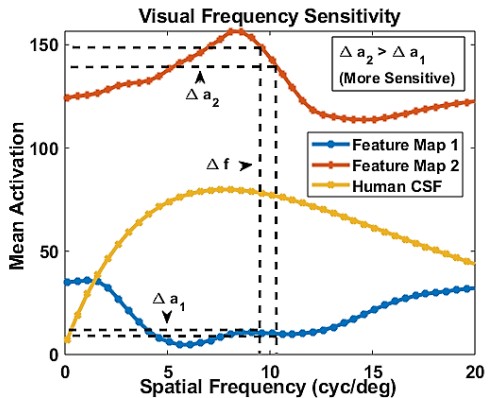

Figure 3: Two different feature maps may have different sensitivities to important visual frequencies.

masking for such a pattern is weak, and distortions are easily perceivable. For complex and irregular image patches, the visual system presents a stronger masking effect.

We have quantified orientation selectivity of different channels in a pre-trained image classification CNN (VGG-16) in Fig. 2.(b). We observe that a significant proportion of channels show a definite orientation selective tuning, such as the ones represented in Fig. 2(b)(a), Fig. 2(b)(b), Fig. 2(b)(j) etc. There channels should in theory be more sensitive in responding to simple patterns. However, quite a few channels show weaker orientation sensitivity such as the ones represented in Fig. 2(b)(c), Fig. 2(b)(k), Fig. 2(b)(n) and Fig. 2(b)(o) etc. We hypothesize the channels that show strong and definite orientation selective tuning, respond better to regular image patterns, which have lower masking thresholds, making these channels deliver better perceptual quality features.

Suppose that within some layer 'k', $a_\theta^m$ be the mean activation of a feature map corresponding to channel 'm' to the input grating of orientation '$\theta$'. Let the maximum mean activation be $a^{\hat{m}} = \max_\theta a_\theta^m$. We model our orientation selectivity attribute for a channel as $\mu_2$ in Eq. (3).

$$\mu_2(k, m) = \sum_\theta (a_\theta^m - a^{\hat{m}})^2 \tag{3}$$

Considering our hypothesis, channels with higher $\mu_2$ should deliver relatively better features of perceptual quality.

## 4.4 PERCEPTUAL EFFICACY SCORE (PE)

Based on our defined attributes, we devise a quantification for the efficacy of channels in pre-trained deep CNNs to deliver good features for perceptual quality, called the Perceptual Efficacy (PE). The perceptual efficacy of a channel with index 'm' in layer 'k' is defined as the product of normalized $\mu_2$ and $\mu_2$.

$$PE(k, m) = \frac{\mu_1(k, m) \cdot \mu_2(k, m)}{\sum_m \mu_1(k, m) \cdot \sum_m \mu_2(k, m)} \tag{4}$$

## 5 EXPERIMENTAL SETUP

We devise an experimental methodology to verify our hypotheses that deep CNN representations that have a higher PE are better perceptual quality features. Let $\mathbb{F}_k$ be the set of all channels within a layer 'k' of a pre-trained CNN (e.g VGG-16).

$$\mathbb{F}_k = \{\Phi_0^k, \Phi_1^k, \ldots, \Phi_M^k\} \tag{5}$$

We constitute subsets of channels from $\mathbb{F}_k$ based on the quantification of our proposed attributes (PE). For example, if there are 128 channels in the *relu2_2* layer of the VGG-16, we can group the top 15% (19 channels) of the total 128 with the highest PE as

$$\mathbb{H}\text{-}15 = \{\Phi_0^k, \Phi_1^k, \ldots, \Phi_m^k\} \tag{6}$$

Similarly, the bottom 15% channels with the lowest PE can be represented as

$$\mathbb{L}\text{-}15 = \{\Phi_0^k, \Phi_1^k, \ldots, \Phi_m^k\} \tag{7}$$

where $\mathbb{H}\text{-}x, \mathbb{L}\text{-}x \subseteq \mathbb{F}_k$ and $x \in (0, 100]$. For $x = 100$, the subsets become the complete set of channels $\mathbb{F}_k$.

To validate our hypotheses, it is necessary to demonstrate that subsets containing higher PE valued channels deliver better perceptual quality features compared to subsets with lower PE valued channels.

### 5.1 Objective Quality Assessment (OQA) Tests

OQA tests correlate the performance of any quality metric, with human subjective assessment and perception of quality (Sheikh et al. (2006)). Human assessment of perceptual image quality is quantified using the Differential Mean Opinion Score (DMOS) over images with varying levels of distortion. DMOS is a quantitative representation of how human observers are able to perceive perceptual differences between natural and distorted images. Metrics that have higher correlation with DMOS scores after regression, measured using statistical indicators such as the RMSE (Root Mean Square Error), LCC (Linear Correlation Coefficient) and the SROCC (Spearman Rank Order Correlation Coefficient), are regarded as better quality metrics (Sheikh et al. (2006)) . The higher the correlation metrics (LCC and SROCC), the better a metric correlates with human ability to perceive perceptual differences.

In our problem setting, we will use Eq. 1 with the different subsets of channels, as defined in Section 5. We demonstrate that for use with Eq. 1, within different CNN layers, channels having higher PE, give much better correlation with DMOS compared to channels with lower PE. Essentially, we demonstrate that CNN channels with our pre-described attributes are indeed better perceptual quallity features.

We use images and DMOS scores from both the LIVE image quality dataset (Sheikh et al. (2006)) and multiple distortion dataset (Jayaraman et al. (2012)) which collectively include images with Gaussian Blur, JPEG compression, JPEG2000, White Noise as well as images which have been corrupted with multiple types of distortions (such as white noise, Gaussian blur and camera noise) simultaneously.

We will repeat our experiment accross multiple layers of several pre-trained image classification CNNs such as AlexNet, ShuffleNet, SqueezeNet and VGG-16.

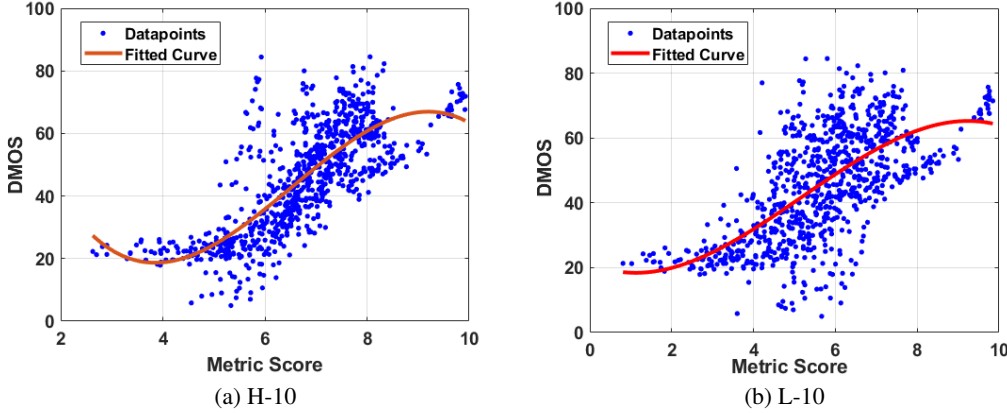

(a) H-10       (b) L-10

Figure 4: Correlation of metric scores in Eq. 1 with human subjective DMOS for the 'fire2_ReLU_exp2x2' layer of the 'SqueezeNet'. It can be seen that the metric in Eq. 1 with the channel subset H-10 has a much better correlation with DMOS, compared to Eq. 1 with the L-10 subset of channels.

Table 1: Objective Quality Assessment Test. The correlation of metric scores delivered by Eq. 1 (for different feature subsets) with human subjective assessment of perceptual quality, quantified by DMOS.

| Network | Layer | Feature Set | RMSE | LCC | SROCC |
|---|---|---|---|---|---|
| VGG-16 | ReLU2_2 | F | 9.8366 | 0.8146 | 0.8028 |
| | | H-10 | 8.8286 | 0.8538 | 0.8486 |
| | | L-10 | 12.3114 | 0.6878 | 0.6806 |
| | | L-90 | 10.5863 | 0.7813 | 0.7739 |
| | ReLU4_1 | F | 9.8149 | 0.8155 | 0.8076 |
| | | H-2 | 8.8183 | 0.8542 | 0.8476 |
| | | L-2 | 10.2338 | 0.7874 | 0.7863 |
| | | L-80 | 9.8485 | 0.8141 | 0.8070 |
| AlexNet | ReLU1 | F | 9.7580 | 0.8179 | 0.8155 |
| | | H-10 | 9.1514 | 0.8419 | 0.8368 |
| | | L-10 | 12.8110 | 0.6553 | 0.6562 |
| | | L-70 | 10.3186 | 0.7936 | 0.7931 |
| | ReLU4 | F | 8.8015 | 0.8548 | 0.8605 |
| | | H-5 | 8.5467 | 0.8637 | 0.8651 |
| | | L-5 | 9.8927 | 0.8122 | 0.8197 |
| | | L-50 | 9.0697 | 0.8450 | 0.8507 |
| SqueezeNet | fire2_ReLU_exp_3x3 | F | 11.2791 | 0.7468 | 0.7397 |
| | | H-10 | 10.8632 | 0.7679 | 0.7625 |
| | | L-10 | 12.6927 | 0.6632 | 0.6614 |
| | | L-50 | 11.6555 | 0.7264 | 0.7199 |
| | fire6_ReLU_exp_3x3 | F | 11.4191 | 0.7394 | 0.7314 |
| | | H-5 | 11.8710 | 0.7142 | 0.7017 |
| | | L-5 | 12.6857 | 0.6637 | 0.6540 |
| | | L-50 | 12.0600 | 0.7063 | 0.6988 |
| ShuffleNet | node7 | F | 11.0810 | 0.7570 | 0.7519 |
| | | H-10 | 9.9055 | 0.8117 | 0.8002 |
| | | L-10 | 14.2481 | 0.5424 | 0.5583 |
| | | L-70 | 11.6409 | 0.7272 | 0.7232 |
| | node17 | F | 9.1354 | 0.8425 | 0.8421 |
| | | H-10 | 8.8577 | 0.8528 | 0.8477 |
| | | L-10 | 11.5070 | 0.7346 | 0.7407 |
| | | L-70 | 9.2306 | 0.8389 | 0.8414 |

Table 2: 2AFC Similarity Test. How well metric decisions conform with human assessment of image triplets .

| SqueezeNet (fire2_ReLU_exp_3x3) | | | | ResNet18 (Res4a_ReLU) | | | |
|---|---|---|---|---|---|---|---|
| F | H-10 | L-10 | L-75 | F | H-2 | L-2 | L-80 |
| 60.23 | 62.85 | 56.08 | 59.83 | 60.69 | 62.53 | 60.10 | 60.21 |
| VGG-16 (ReLU3_2) | | | | AlexNet (ReLU4) | | | |
| F | H-5 | L-5 | L-50 | F | H-2 | L-2 | L-75 |
| 59.97 | 60.86 | 58.28 | 59.41 | 64.62 | 63.38 | 61.30 | 62.35 |

## 5.2 2AFC SIMILARITY TESTS

In the 2AFC test, two distorted images are shown to an observer and he/she is asked to rate which one is closer to the ground truth, in perceptual appearance. This process is repeated for multiple image triplets and observers per-triplet to construct a data-set called the Berkley-Adobe Perceptual Patch Similarity Data-set (BAPPS) (Zhang et al. (2018)). Objective metrics such as the one in Eq. 1 are thereafter evaluated to see how well they conform to the pair-wise human judgment. For example, in an image triplet, let $x_0$ and $x_1$ be two distorted versions of the ground truth image $x_g$ that are shown to 5 human observers, 4 of which judge $x_0$ to be closer to $x_g$, as opposed to $x_1$ being closer to $x_g$. If an objective metric evaluates $x_0$ to be closer to the ground truth, it will get an 80% credit which in the opposite case would be 20%.

The BAPPS data-set contains images with distortions such as super-resolution, frame-interpolation and deblurring, which do not have subjective DMOS data-sets available online. Therefore, as a secondary experiment, we perform a 2AFC test on super-resolution, frame-interpolation and video-deblurred frame images in the BAPPS data-set with Eq. 1 for different channel subsets defined in Section. 5. In order to verify our hypothesis, we will show that subsets that contain channels with higher PE, deliver better perceptual quality features.

## 6 RESULTS AND DISCUSSIONS

Table 1 quantifies the correlation of Eq. 1 with DMOS for different subsets of channels, constructed on the basis of our described attributes, as explained in Section 5. Table 1 validates our hypothesis that within a CNN layer, channels which have higher PE (Eq. 4) deliver better perceptual quality features. It must be reminded that higher LCC and SROCC indicate better correlation of Eq. 1 with human ability to discern perceptual differences (DMOS). It can be observed that very small proportions (2%-10% of total) of channels with the highest PE, deliver better perceptual quality features compared to a much higher proportion (50%-90% of total) of channels having lower PE. Furthermore, in a majority of cases, a small proportion of channels that have our described attributes (higher PE), perform even better than the complete set of channels in the layer. This implies that our proposed attributes are indeed important characteristics that make learned deep CNN representations good perceptual quality features.

Table. 2 shows the results of our secondary 2AFC similarity test on the super-resolution, frame-interpolation and video-deblurring distorted images in the BAPPS data-set. It can been seen that yet again, similar to the conclusion in the primary QQA experiment, the subsets with channels having higher PE are better perceptual quality features compared to even much larger subsets having channels with lower PE.

## 7 FUTURE WORK

We have proposed a model to explain and interpret which channels in pre-trained image classification CNNs deliver better perceptual quality features. The model may be used to improve the use of deep representations as perceptual quality features by helping in feature selection for IQA methods such as (Bosse et al. (2017)) and maybe designing channel attentive mechanism to improve the perceptual loss (Johnson et al. (2016)). The model may also be reference for learning better perceptual quality feature representations which may benefit a wide variety of applications. Furthermore, the model may be enhanced to include more psychophysical factors such as eccentricity etc. Another possible application may be CNN-based image compression where prior knowledge of the potential efficacy of different channels may help efficient perceptual compression of redundant image data.

## 8 CONCLUSIONS

Deep CNN representations of pre-trained image classifications CNNs have been popularly used as perceptual quality features for perception orientated applications such as CNN based quality assessment, image/video super-resolution and many image-to-image translation problems. In this paper, as a novel and first effort, we have linked basic human visual perception models to pre-trained deep CNN representations in order to explain and interpret them as perceptual quality features. Based on

masking characteristics in human visual perception, we formulate attributes of channels in different layers of pre-trained networks, and experimentally demonstrate that the attributes are important characteristics that make some deep CNN representations better perceptual quality features compared to others.

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

## A  GRATING GENERATION

In this section, we present details behind the generation of sinusoidal gratings of different spatial frequencies.

The contrast sensitivity function is expressed on the domain of spatial frequency in cycles per degree (cyc/deg). The cycles per degree express the number of sine cycles captured by the observer per unit degree of observation. Obviously, the distance of viewing and dimensions of the screen play an important part in this measurement.

We essentially generate gratings in the computer simulation in cycles per pixel. Let the display screen being used in the experiment have a height 'h' inches and resolution 'r' pixels per inch. The optimal viewing distance in psychovisual experiments should satisfy a function called the PVD (42 (2002)). The PVD is a function that expresses the optimal ratio of viewing distance to the height of the display screen. The optimal viewing distance 'd' for the screen with height 'h' can be calculated using the PVD.

The transformation between cycles/degree and cycles/pixel is

$$\frac{cycles}{pixel} = \frac{cycles}{degree} \times \frac{degrees}{pixel} \tag{8}$$

Where

$$\frac{pixels}{degree} = \frac{180}{\pi \times d \times r}$$ (9)

Therefore,

$$\frac{cycles}{pixel} = \frac{cycles}{degree} \times \frac{\pi \times d \times r}{180}$$ (10)

We have tested with a number of different display systems of SD, 2K and 4K resolutions. Considering that the PVD takes the viewing angle into account, the changes in the resultant spatial frequencies of the gratings are small and insignificant. Therefore, it can be concluded that the choice of display system has a negligible effect on the experiment.

For the generation of grating with fixed spatial frequency and varying orientation, the experimental setup is the same.

## B    ADDITIONAL NETWORKS

We demonstrate the validity of our hypothesis for other pre-trained image classification CNNs as well. These networks include:

- GoogleNet (Szegedy et al. (2014))
- MobileNet-v2 (Sandler et al. (2018))
- ResNet-18 (He et al. (2015))

It can be seen in Table 3 that our hypothesis regarding important attributes is valid for these additional CNNs as well. Small proportions of channels (H-(5-10)) with higher PE (Eq. 4) deliver much better perceptual quality features compared to a much higher proportion of channels with lower PE (H-(45-80)) and even the complete set of channels in the layer (F). A scatter plot of some correlations, shown side by side is depicted in Fig. 5.

## C    ADDITIONAL EXPERIMENT

We have included an additional experiment with the aim to demonstrate that our work and experimental analysis techniques (Objective Quality Experiments) are generalizable to practical imaging problems. The experiment will be an x4 CNN based super-resolution task. However, a little background on the perception-distortion trade-off is necessary before proceeding.

### C.1    PERCEPTION-DISTORTION TRADE-OFF

It is a known fact that traditional distortion metrics such as PSNR and SSIM are not well correlated with perceptual quality and the distortion measured by PSNR and SSIM for a degraded image indicates the net deviation from its reference image. Many different images of different perceptual qualities may have a same distortion with respect given reference. Perceptual quality, on the other hand, is a no-reference quantification of how natural an image appears (conformity with natural image statistics). This is why objective distortion measures such as PSNR and SSIM do not necessarily account for perceptual quality. Keeping in view the recent interest in perception-oriented imaging applications, to quantify no-reference perceptual quality, a combination of the metrics such as NIQE (Mittal et al. (2013)) and NRQM (Ma et al. (2016)) has been used as a perceptual indicator (PI) in recent works (Blau et al. (2018)). The lower the Perceptual Index (Eq.11), the better the perceptual quality of an image.

$$PI(I_{in}) = \frac{1}{2}((10 - NRQM(I_{in})) + NIQE(I_{in}))$$ (11)

The work of (Blau & Michaeli (2018)) demonstrates that distortion and perceptual quality are in a trade-off relation and this trade-off is the correct measure for quantifying the perceptual efficacy of image restoration algorithms as explained in (Blau et al. (2018)). Consider Eq.(12), a standard

Table 3: Objective Quality Assessment Test. The correlation of metric scores delivered by Eq. 1 (for different feature subsets) with human subjective assessment of perceptual quality, quantified by DMOS.

| Network | Layer | Feature Set | RMSE | LCC | SROCC |
|---|---|---|---|---|---|
| GoogleNet | conv2_ReLU_3x3 | F | 9.2730 | 0.8370 | 0.8351 |
| | | H-5 | 9.1360 | 0.8425 | 0.8364 |
| | | L-5 | 12.6595 | 0.6654 | 0.6674 |
| | | L-80 | 9.6636 | 0.8218 | 0.8203 |
| | inception_4a-ReLU_3x3 | F | 10.2264 | 0.7977 | 0.8061 |
| | | H-5 | 9.8592 | 0.8137 | 0.8201 |
| | | L-5 | 10.8882 | 0.7667 | 0.7750 |
| | | L-45 | 10.0326 | 0.8063 | 0.8163 |
| MobileNet-v2 | block1_expand_ReLU | F | 11.9441 | 0.7099 | 0.7017 |
| | | H-10 | 11.6059 | 0.7292 | 0.7256 |
| | | L-10 | 13.7130 | 0.5884 | 0.5825 |
| | | L-70 | 12.7912 | 0.6566 | 0.6505 |
| | block3_expand_ReLU | F | 10.1957 | 0.7991 | 0.8063 |
| | | H-10 | 9.2423 | 0.8385 | 0.8459 |
| | | L-10 | 13.2810 | 0.6219 | 0.6223 |
| | | L-70 | 10.7877 | 0.7716 | 0.7804 |
| ResNet-18 | Res2a_ReLU | F | 10.8622 | 0.7680 | 0.7702 |
| | | H-10 | 10.0841 | 0.8040 | 0.7898 |
| | | L-10 | 11.6195 | 0.7284 | 0.7339 |
| | | L-75 | 11.2807 | 0.7467 | 0.7549 |
| | Res4a_ReLU | F | 9.1073 | 0.8436 | 0.8611 |
| | | H-5 | 9.2559 | 0.8379 | 0.8509 |
| | | L-5 | 10.1132 | 0.8028 | 0.8072 |
| | | L-75 | 9.3484 | 0.8344 | 0.8518 |

loss function for CNN based SR, which is defined as a combination of a pixel-wise loss ($l_1$) and the perceptual loss ($l_p$).

$$L_p = \alpha \cdot l_1 + (1 - \alpha) \cdot l_p \tag{12}$$

Increasing $\alpha$ in Eq.(12) will lead to a decrease in distortion (increasing SSIM), which according to the perception-distortion trade-off, should result in images with lower perceptual quality (higher PI). Similarly, decreasing $\alpha$ will give more weight to the perceptual loss, which would result in images with better perceptual quality (lower PI), but lower SSIM.

## C.2 X4 SUPER-RESOLUTION EXPERIMENT SETUP

In this experiment, we have used the well known VDSR (Kim et al. (2016)) network trained on images from the DIV2K data-set. The network was separately trained on two loss functions. The first being the standard perceptual loss in Eq.(12) and the second a slight modification (Eq.13) based on our analysis.

$$L_{pv} = \alpha \cdot l_1 + (1 - \alpha) \cdot l_{pv} \tag{13}$$

where

$$l_{pv} = \frac{1}{M \cdot W \cdot H} \sum_{m=1}^{M} w_m^k \|\Phi_m^k(I_{out}) - \Phi_m^k(I_{GT})\|_2^2 \tag{14}$$

and

$$w_m^k = \frac{\mu_1(k, m)}{\sum_m \mu_1(k, m)} \tag{15}$$

Eq.(13) presents a version of a perceptual loss in which the channels are weighted (Eq.14) as per our defined visual frequency sensitivity (theoretical importance of a channel based a defined attribute). If our hypothesis is correct, the loss function in Eq.(13) should be able to perform better compared to Eq.(12) in terms of providing a better perception-distortion trade-off for super-resolution. The results should consequently serve as an additional experiment to reinforce our hypothesis and

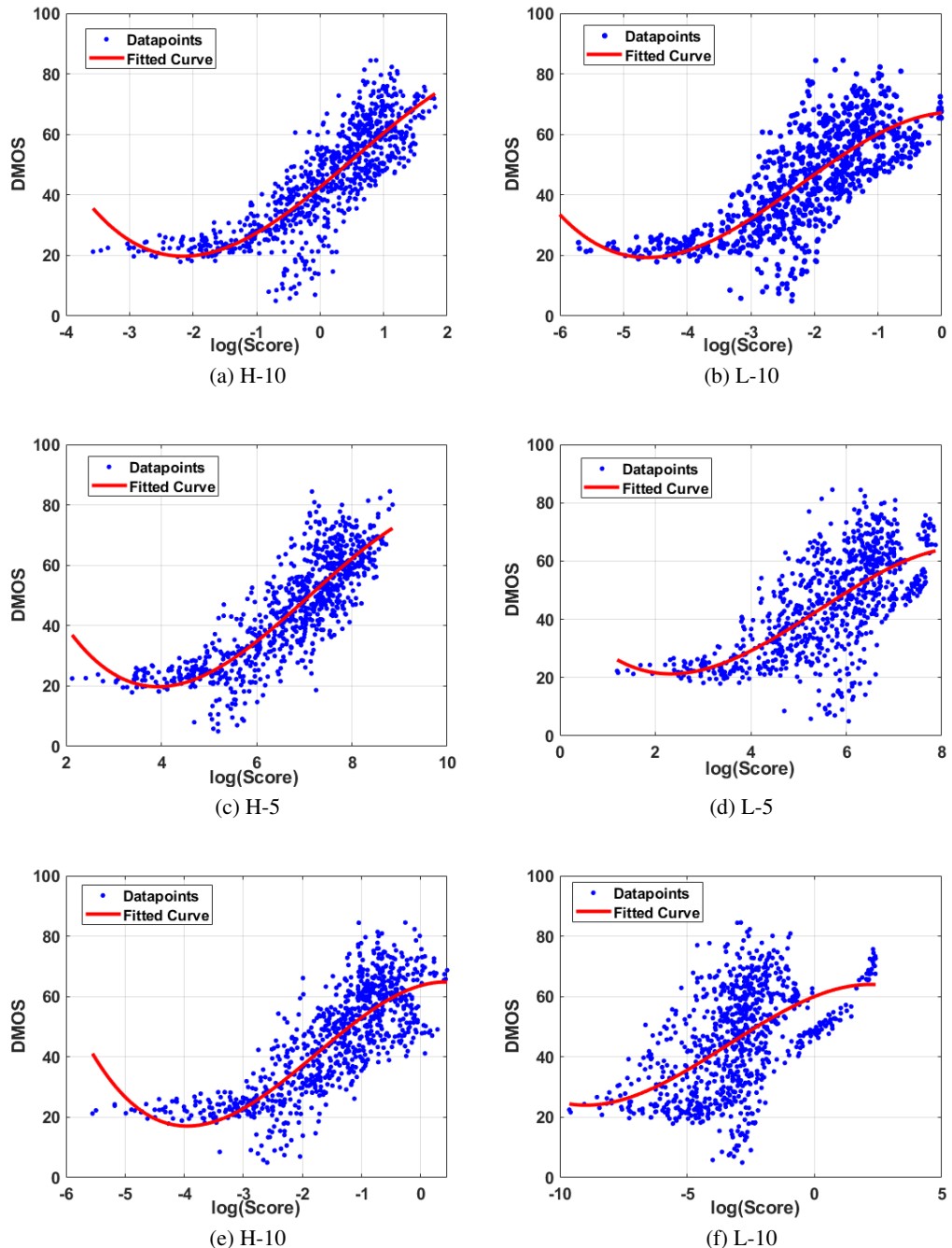

Figure 5: Correlation of metric scores in Eq. 1 with human subjective DMOS shows that the metric in Eq. 1 with the channel subset H has a much better correlation with DMOS compared to Eq. 1 with the L subset of channels. Each pair shown side by side is for a different network.

demonstrate the transfer-ability of objective quality assessment experiment conclusions to practical applications. The results should also demonstrate the ability of an independent attribute as an indicator of the perceptual abilities of pre-trained channels. The results are not be intended to deliver state of the art results, just to verify the generalization of our analysis to practical problems.

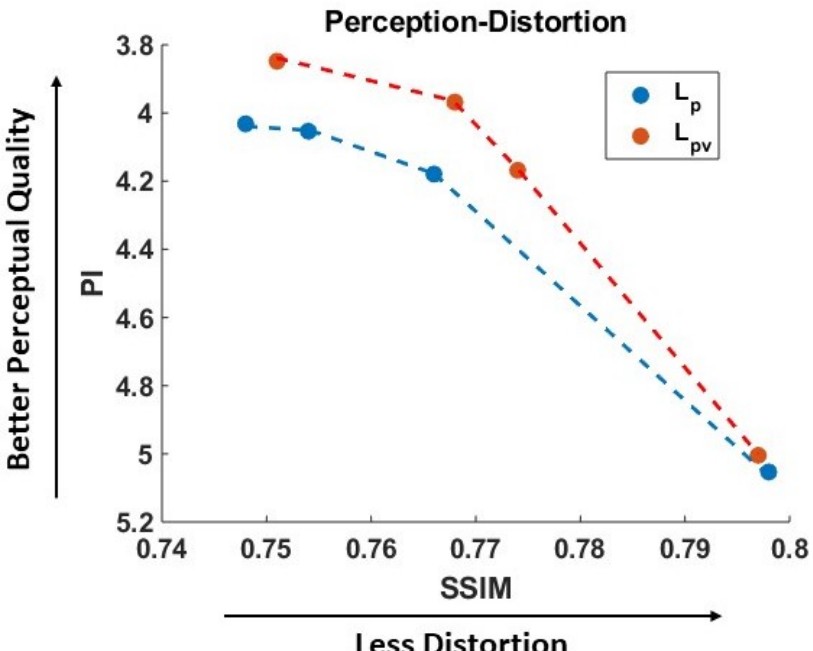

Figure 6: (Eq. 13) improves the perception-distortion trade-off compared to the perceptual loss (Eq. 12) when used for training with different $\alpha$ settings for an x4 SR with the VDSR on the DIV2K data-set.

## C.3   RESULTS

The *ReLU4_2* layer of a pre-trained VGG-16 was used for perceptual loss terms in Eq.(12) and Eq.(13). The experiment was repeated for different $\alpha$ values to analyze the perception-distortion trade-off delivered by both loss functions. The SSIM and PI reported in Fig.6 are averaged over all the 100 images in the test-set.

It can clearly be seen in Fig.6 that $L_{pv}$ in Eq.(13) improves the perception distortion trade-off compared to $L_p$ in Eq.(12). The improvement is evident from the fact that the perception-distortion curve delivered by $L_{pv}$ is higher compared to the one delivered by $L_p$ for various $\alpha$ (different points on the curve correspond to different $\alpha$ in Eq.(12) and Eq.(13)). Therefore, at the same level of distortion (SSIM), $L_{pv}$ is able to generate images with better perceptual quality compared to $L_p$.

The result reinforces the validity of the experimental results presented in the main paper and demonstrate that they have consequence for practical applications such as super-resolution. The results of this experiment also iterate the fact that conclusions derived from Objective Quality Assessment(OQA) experiments have a consequence for practical image generation problems.

