# OpenReview forum: "Analysis and Interpretation of Deep CNN Representations as Perceptual Quality Features"
_ICLR.cc/2020/Conference — Reject_

### Official Review · AnonReviewer3 · 2019-10-14
**Official Blind Review #3**

**Rating:** 6

**Review:**

In the article the authors propose to measure quality of CNN-features by quantifying the orientation tuning and spatial frequency sensitivity of the features. The underlying hypothesis is that properties of features in the human visual cortex are also indicators for quality in CNNs. The authors devise an experiment similar to experiments performed on mammals to check which features are active under which types of basic patterns. Afterwards, a loss-function is devised that uses proportions of the best or worst features according to the metrics and it is shown that features that have high values on the metrics also lead to good performance.
-----------------------------------------------------------------------------------------
I have a problem understanding some of the metrics used. In the introduction, the following claim is made:
"The first attribute is sensitivity to spatial frequencies at which there is minimal contrast masking in human visual perception". To my understanding, the metric to measure this is (2) in 4.2. a_m^k is to my understanding the average response of the feature when given an image with orientation-frequency f. therefore, the derivative should be "the change of activation under change of frequency". I feel unable to connect this with the initial hypothesis, as it does not mention change of frequencies. i would have expected the correlation between CSF(f) and a^k_m(f),
did you have a reason why you did not chose correlation?

Similarly, for mu_2 in (3) you are using the maximum value. Is there a reason not to use the mean? in that case mu_2 would be the variance, which would be a natural measure for orientation selectivity.

------------------
Is there a way to make Table 1 more pleasing for the human eye wrt the discussion of the results?

**Experience Assessment:**

I have read many papers in this area.

**Review Assessment: Checking Correctness Of Derivations And Theory:**

I assessed the sensibility of the derivations and theory.

**Review Assessment: Checking Correctness Of Experiments:**

I assessed the sensibility of the experiments.

**Review Assessment: Thoroughness In Paper Reading:**

I made a quick assessment of this paper.

---

> ### Author Response · Authors · 2019-11-06
> **Thank you Reviewer # 3**
>
> Thank you very much for taking the time to review our manuscript and providing a thorough review. You have raised some interesting questions. The prime focus of this paper is to link basic human visual perception to interpret the ability of pre-trained image classification CNN channels as perceptual quality features, a problem that has never been addressed.
>
> Firstly, let us explain why we have employed the concept of sensitivity to important (low contrast masking, high CSF valued) spatial frequencies rather than using correlation, which at first glance seems a more logical choice. What needs to be remembered is that the perceptual loss is a differential metric i.e It is designed to quantify perceptual differences between two images. Therefore, what should actually be preferred is not absolute activation, rather the activation should changes with perceptually important distortions (because the difference in activation between distorted and ground truth is the thing we are interested in). For example, lets suppose two CNN channel's have an all pass type response, would they be good for the perceptual loss? The answer is no, because they will not change their activation much according to the distortion on the input image, relative to the ground truth. What should be preferred is a sharp change in activation at high CSF valued spatial frequencies, so that the CNN channels are able to respond much better to perceptually important distortions, COMPARED to unaffected ground truth images.
>
> Secondly, regarding our quantification of orientation selectivity and why we have not used the mean. Suppose for a random data stream, one may take the mean and variance, which will give us a distribution. The orientation selectivity of channels is actually somewhat analogous to a distribution (a peak at some orientation and a fall-off). Orientation selectivity, as per our definition is how selective a channel is to a particular orientation (the peak). Therefore, In order to maximize selectivity, we prefer a fast fall-off in tuning around the peak. The formulation in the manuscript rewards higher selectivity of a particular orientation (the peak orientation) wrt to all other orientations, which is what we desire. If we were to use the mean, it would compute the spread wrt to the average orientation selectivity over all orientations, which is not what we are after.
>
> Regarding Table. 1, I understand the a jargon of number is often difficult to understand, plus its not visually pleasing. However, the experimental techniques and reported evaluation metrics are the standard to analyze perceptual quality metrics. Considering that we analyzed multiple layers of multiple network using three evaluation metrics, graphical representations would have been very difficult. I apologize for the inconvenience. I have added additional explanation in the results section to help readers understand the table better, I hope that helps.
>
> Furthermore, we have added an additional experiment in the Appendix (Section. C) that demonstrates that our results are also consequential for practical imaging problems such as SR, reinforcing our analysis. I hope you like it. Thank you very again for taking the time to review our manuscript and providing a thorough review, we are at your disposal for any more queries that you may have.

---

> > ### Comment · AnonReviewer3 · 2019-11-07
> > **Still unsure about first metric**
> >
> > Hi,
> >
> > Thanks for your thorough explanation. I accept the second metric now as is. I think i might have misunderstood "selectivity" in this context.
> >
> > I am still unsure about the first metric and I did some reading in the meantime as well. You are writing the following in your text:
> >
> > "The CSF depicts human ability to perceive contrast changes as a function of spatial frequency. The spatial
> > frequencies where the CSF has a higher value, correspond to lower contrast masking thresholds in
> > perception.[...] Our hypothesis is that the deep representations that are more sensitive to high CSF valued spatial frequencies, can be better features of perceptual quality"
> >
> > From my understanding of the literature, contrast masking refers to sensitivity towards changes of amplitudes in frequency domain, i.e. how sensitive human perception is to changes of amplitudes at a given frequency. e.g. compare to
> >
> > Gordon E. Legge and John M. Foley, "Contrast masking in human vision," J. Opt. Soc. Am. 70, 1458-1471 (1980)
> >
> > here contrast masking threshold were measured by presenting the same sine-grating at different Amplitudes and there was no change of frequency involved, so I don't see why you can use da/df here, which measures change of response under change of frequency. Instead, what i could see is the use of da/dA, where A is the amplitude of the grating at frequency f.

---

> > > ### Author Response · Authors · 2019-11-07
> > > **More on the first metric.**
> > >
> > > Thank you very much for your continued interest in our work. I think I understand where you are coming from. I apologize if some things in the following explanations seem simple and repetitive, as you probably already know them, but I just wanted to give a brief picture. You are actually referring to psychovisual experiments that are used to measure the human contrast sensitivity function. The basis of human contrast sensitivity is lateral inhibition, which refers to the tendency of a neuron to change its activation with respect to neighboring neurons, hence the concept of relative perception of intensity in human vision.
> > >
> > > The procedure you are referring to is the one which is used to derive the contrast sensitivity function. The human observer sits in-front of a screen and is shown a sinusoidal grating of a fixed spatial frequency with a specific contrast. You have referred to this as amplitude, which i think should be contrast (the intensity difference between bright and dark bars etc). Now, the contrast of the grating is steadily decreased till the contrast at which the observer is unable to perceive any contrast (it seems like a grayish plain pattern). This gives us a contrast threshold AT A PARTICULAR FIXED SPATIAL FREQUENCY. This process is repeated again and again for different spatial frequencies, giving us a a curve of contrast thresholds versus spatial frequencies. The CSF is an inverse of this curve i.e the spatial frequency at which there was a minimum contrast threshold (observer was able to perceive even a very small contrast) corresponds to the maximum of the CSF (its about 5.7 c/d for a normal aged human). Therefore, the contrast sensitivity function comes out to be a representation of human visual acuity (our ability to resolve detail) as a function of spatial frequency. This is the standard experimental procedure neuro-scientists use to measure the human CSF, which has been measured many times and is available as a mathematical function . Lets now proceed to our experiment.
> > >
> > > In our experimental method, we input sinusoidal gratings (fixed contrast) of increasing spatial frequency into the CNN one by one, thus giving us the ability to get an activation vs spatial frequency response curve for each channel. This in no way should be thought of as a CSF for the CNN. What we are after is a tuning curve which tells us about affinity of channels to different spatial frequencies. Now in the formulation of our first attribute, the CSF (which we already have, courtesy of neuro-scientists) is an importance weighting of different spatial frequencies (weighted sum).
> > >
> > > Suppose you have a natural image and it is distorted by some blur etc, input into the pre-trained CNNs. The noise will effect the underlying spatial frequencies in the natural image. In order for a feature map to be an effective perceptual quality feature, it should respond strongly to the change (between natural and distorted images) in important underlying spatial frequencies. The gradient da/df tells us how sensitive a channel is to a particular spatial frequency 'f', meaning how much its activation is expected to change considering perturbation on frequency components corresponding to the frequency 'f' (like a gradient of the tuning curve at the frequency). The CSF is an importance weighting that tells our metric which spatial frequencies are more important (more likely to have an effect in the perception of even the slightest distortion). So in essence, what we are looking for are channels that should have stronger sensitivity to pick up distortions effecting important spatial frequencies. The term 'pick up' refers to a change in activation (difference between the channel activation's for distorted image and natural image) on the neurons in the feature map. As the perceptual loss is used to QUANTIFY PERCEPTUAL DIFFERENCES between two images, these changes in activation for each channel in the loss function, are what actually accumulate into what contributes in the ability of the loss to quantify of distortions. In summary, If the channel responds better to more perceptually important distortions, it delivers better perceptual quality features, as we have verified using the OQA experiments as well as the newly added SR experiment (Appendix section C).
> > >
> > > Thank you very much again for your continued interest, I am again at your disposal for any more queries.

---

### Official Review · AnonReviewer2 · 2019-10-23
**Official Blind Review #2**

**Rating:** 3

**Review:**

I thank the authors for their detailed response. Some of my questions have been addressed in the rebuttal, but as far as I can tell very few modifications have been made to the paper: mainly an additional super-resolution experiment in the appendix, which goes in a good direction, but currently comes across as quite preliminary. So I think the paper still has most of the weaknesses I mentioned and thus it is not quite fit for publication. But I do encourage the authors to strengthen the experiments (for instance by addressing the points I raised, or by some other means) and resubmit elsewhere.

---

The paper proposes an approach to analyzing the properties of "perceptual metrics" used in deep learning image generation methods. "Perceptual metrics" are computed by measuring distances between images not in the pixel space, but i na feature space of a pre-trained cNN. The proposed analysis method, inspired by studies of human perception, is based on measuring the response of these features to sinusoidal gratings of varying frequency or orientation. Based on these responses, the paper proposes a "Perceptual Efficacy Score" that should measure the importance of certain feature in the feature maps for the performance of a perceptual metric. Experiments show that indeed distances measured between features with high score better correlate with human judgement of image similarity than distances between features with a lower score.

I find the paper quite interesting, but lean towards rejection at this point. This i smainly because the experiments seem somewhat anecdotal and incomplete, see below for further details.

Pros:
1) Application of methods from psychology/neuroscience to artificial neural networks is an interesting avenue of work. Moreover, better unsderstanding of "perceptual metrics" is of wide interest for various image processing applications.
2) The proposed score seems to indeed correlate quite well with the importance of features for human judgement of image similarity.
3) Presentation is mainly clear.

Cons:
1) Experiments are not very exhaustive and at times a bit confusing. For instance:
1a) Results are sometimes presented in a confusing way. In Figure 4 first of all it is not quite clear what points correspond to I guess each point is an image) and, second, it is not very obvious that the correlation is higher i none of the plots. In tables 1 and 2 it is confusing that different percentiles for H and L are used for different networks/layers. Is this based on some tuning? Then the tuning process should be clearly explained. Moreover, it might be useful to report the full curves of performance as a function of the percentage of features used.
1b) There are no baselines and there is not much justification of computing the "Perceptual Efficacy" score the way it is computed. What if one uses only the orientation-based score? Or only the frequency-based? What if one selects the most relevant features in a data-driven way (based on correlation on a training set)? What if one selects subsets of features randomly?
1c) While the method is inspired by methods used for studying natural vision systems, there is no connection to human experiments. It would be interesting to see a comparison of frequency and orientation tuning of features in a CNN to human cells (as I understand, the latter should be available in prior works?).
1d) It would be great to see the selected features be used not only for offline image similarity assessment, but also for training image processing models - in the end, this has been the main use of "perceptual metrics". Do they lead to improved results?
1e) Since the paper is about (subjective) image quality, it might be useful to show some qualitative results, potentially in the appendix if space is an issue.

2) There are some issues with the presentation:
2a) I had a hard time understanding what exactly "Contrast Sensitivity Function" and "contrast masking" are.
2b) Minor issues:
- In the abstract: "trained object detection deep CNNs" - I guess image classification is meant
- Beginning of Section 2: "Section. 2", "convolution layer as collection channels"
- Section 4: "corresponds the the peak:
- Section 5.2 "Berkeley-Adobpe"

**Experience Assessment:**

I have published one or two papers in this area.

**Review Assessment: Checking Correctness Of Derivations And Theory:**

I assessed the sensibility of the derivations and theory.

**Review Assessment: Checking Correctness Of Experiments:**

I carefully checked the experiments.

**Review Assessment: Thoroughness In Paper Reading:**

I read the paper at least twice and used my best judgement in assessing the paper.

---

> ### Author Response · Authors · 2019-11-06
> **Thank you Reviewer # 2 (Reply 1/2)**
>
> Thank you for taking the time to read our manuscript and providing a thorough review. The prime focus of this paper is to link basic human visual perception to interpret the ability of pre-trained image classification CNN channels as perceptual quality features, a problem that has never been addressed. I would try my best to address your concerns.
>
> Regarding interpreting Fig. 4 visually, the ideal quality metric would be on which all the points lie on a straight line. Each point does indeed represent an Image, the y-axs being a score of how humans evaluated it in comparison to the ground truth, the x-axs being how the metric evaluated it. What we do not want is images  with multiple levels of distortions giving the same metric score, and vice versa, Image with the same level of distortion giving varying metric scores. If you compare the image in Fig. 4, you can observe that with the high PE result, the vertical spread of points wrt to constant metric scores is reduced, which means that images with the same metric scores are now less variant in perceptual quality, and vice versa. Also, kindly refer to Fig. 5 in the appendix for further results. If you prefer, I can replace Fig. 4 with a plot that is more interpretable.
>
> Regarding the different percentiles in Table. 1 and 2. They were not due to any tuning characteristics or intrinsic properties of channels. The different values were chosen to see that the results are valid for a wide variety of divisions, also it is important to remember that all layers do not have the same number of channels i.e a 5% subset from a layer with 512 channels makes sense, not so much for a layer with 32 or 64 channels. They divisions are in no way biased or influenced by some prior.
>
> Regarding the issue of relation to human experiments, we actually have human perceptual input in our experiments, which exists in the form of DMOS (human quality scores), derived from a population of human subjects. The CSF is actually a representation of overall human perception of visual stimulus, which is a result of a combination of different levels of neural activation's. As the problem scope is towards overall psychophysics and PERCEPTION OF DISTORTIONS, not individual activation's on a neuron level, that route was not very helpful here, but it is a very good idea for some other study, more focused in understanding CNN activation similarities with the visual cortex. To the best of our knowledge, you will not find any prior work on the problem and approach we have adopted.
>
> Regarding training for image processing models, an ADDITIONAL EXPERIMENT has been added in the appendix, which demonstrates that our analysis is consequential for practical imaging problems such as super-resolution. That result also further reinforces the validity of hypothesis are techniques. The additional experiment hopefully alleviates some of your concerns regarding the experiments.
>
> Furthermore, the minor issues have been mitigated with additional explanations and corrections.
>
> I hope I have sufficiently addressed your major concerns. Thank you again for your time and thorough review, we are at your disposal for any more queries.

---

> > ### Author Response · Authors · 2019-11-14
> > **Thank you Reviewer # 2 (Reply 2/2)**
> >
> > Regarding the baselines and the justification of computing the "Perceptual Efficacy". The main point is that the PE is aimed as a quantification to demonstrate that our proposed attributes are important characteristics for deep CNN channels that deliver good perceptual quality features. The specific problem we have addressed has no precedence, so we had to design the experiments on our own. The PE as defined in the manuscript is the simplest model that gives equal importance to each attribute, like an equi-weighted combination. Furthermore, we have sufficiently demonstrated (making use of two different experiments of separate data-sets, using multiple networks and layers) that it is possible to discriminate between CNN channels within layers on the basis of perceptual attributes.
> >
> > Your first query hints towards a question of how well exactly does one attribute performs in comparison to another, the answer to which would lead to exponent weighting for relative importance of attributes in the PE equation. The scope of this paper was to formulate human perception inspired attributes for CNN channels and demonstrate that they can interpret the efficacy of pre-trained channels to deliver perceptual quality features. Exactly how good one attribute is compared to another would require significant data analysis, which is not in the scope of this paper. However, considering your concern, I have added an additional experiment in the Appendix (section C) which demonstrates that a single attribute independently also does really well, for a practical problem (super-resolution).
> >
> >  Regarding a "Data-driven" approach to compute PE, something like using regression to learn channel importance weights etc. While this approach might give you channel efficacy scores that work well for some experiments, it would do absolutely nothing in improving our understanding and interpreting what makes the representations good perceptual quality features, which is the prime focus of the paper. We have adopted a bottom-up approach, in line with the scientific method where we develop a hypothesis, inspired by a natural mechanism, and demonstrate its validity, not first using some data-driven methods to LEARN scores and then trying different things to understand what they mean, which would defeat the purpose. Furthermore, we have sufficiently demonstrated that even a very small proportion of good channels is better than the entire set in the layer, therefore it will most definitely be better then a random selection (which again would do nothing to help interpretation).
> >
> > The problem statement we have posted in our manuscript states "Are all of the channels equally effective in delivering good perceptual quality features? Are some channels better than others and if so, what attributes make them better?". I believe we have sufficiently addressed the scope of this problem.
> >
> > I hope I have sufficiently addressed your major concerns. Thank you again for your time and thorough review, we are at your disposal for any more queries.

---

### Official Review · AnonReviewer1 · 2019-10-27
**Official Blind Review #1**

**Rating:** 3

**Review:**

This paper proposes an analysis of convolutional neural networks (CNNs) features the basis for making perceptual quality comparisons. The analysis is based on the proposed Perceptual Efficacy (PE) Score that measures spatial frequency and orientation selectivity of CNN features. The hypothesis put forward by the authors is that a CNN features with high PE score can be used to formulate a perceptual loss (Eq. 1) that correlates well with human image quality judgement. The authors use a dataset of human image quality judgements to assess their hypothesis.

One issue I see with the hypothesis as stated is that in the definition of frequency selective features, the authors make use of
the Contrast Sensitivity Function (CSF) which quantifies the dependency of human perceptual characteristics on frequency. So in the definition of the PE score we have embedded knowledge of human perceptual sensitivity. Is it therefore surprising  that we see correlation between the high PE features and human judgements of quality?

Experimental results: The scatterplot presented in Figure 4 does not say to me what the authors claim it should. I do not see a significant difference between the low-PE features and the high-PE features in terms of their correlation with human image quality judgement (as measures in this case by the DMOS). I also find the large table of number in Table 1 to be rather
impenetrable. I would recommend an alternative method of presentation to make the desired point.

Clarity: There are many undefined terms and acronyms (eg. SISR, HVS are not defined, while DMOS and SROCC are not described). Also, the description of visual masking in Sec. 4.3 was confusing and difficult to follow. Otherwise the writing was reasonably clear.

Impact and significance: Overall, the findings of the paper are not terribly surprising and as discuss above, given the use of the CSF (quantifying human perceptual characteristics) in the definition of the CNN Perceptual Efficacy (PE) score, it would seem rather surprising that a correlations would not be found. As a result of this as well as the rather narrow nature of the study involved, I am inclined to think that the impact potential of this paper would be rather low.

**Experience Assessment:**

I have read many papers in this area.

**Review Assessment: Checking Correctness Of Derivations And Theory:**

I carefully checked the derivations and theory.

**Review Assessment: Checking Correctness Of Experiments:**

I assessed the sensibility of the experiments.

**Review Assessment: Thoroughness In Paper Reading:**

I read the paper thoroughly.

---

> ### Author Response · Authors · 2019-11-06
> **Thank you Reviewer # 1**
>
> Thank you for taking the time to read our manuscript and providing a thorough review. The prime focus of this paper is to link basic human visual perception to interpret the ability of pre-trained image classification CNN channels as perceptual quality features. The PROBLEM STATEMENT we have posted in our manuscript states "Are all of the channels equally effective in delivering good perceptual quality features? Are some channels better than others and if so, what attributes make them better?", a problem that has never been addressed.
>
> We are pleased to know that you think our conclusions are not surprising. Quite a majority of the Deep Learning community is still resistant to works that provide a human vision based perspective to neural networks. Our main approach has been to link basic human visual characteristics such as frequency sensitivity and orientation selectivity to interpret deep CNN representations as perceptual quality features. The result may not be that surprising, it might be quite intuitive  (considering what we know about human visual perception), which is good from an interpretation point of view, but we believe it is not inconsequential as it adequately address the unaddressed problem statement we have put forward. Firstly, linking human visual perception characteristics to CNNs has been a bottleneck (as we have explained). Only after our novel approach of using gratings to quantify frequency and orientation selectivity of CNN channels, it becomes possible to apply our knowledge of perception (CSF etc) to design models to explain and interpret them.  Secondly, despite rapid progress towards new and complex DL methods, the interpretation of CNN models has not received much attention, we are moving rapidly into a direction where we know less and less about what is actually going on. It is important from a scientific perspective, to understand and interpret learned representations, which is also a major focus of the ICLR. It is quite interesting that learned representations which have the prime task of classifying images, deliver remarkable perceptual quality features, and we can use human vision perception to get insight on why. This is the first work that dives into the why and how, from a perspective of visual perception. Thirdly, we have added an ADDITIONAL EXPERIMENT in the appendix (section C) to demonstrate how our analysis is actually consequential for practical perception-oriented imaging problems, such as super-resolution. As we have explained, understanding perceptual aspects of deep representations can also impact perception-oriented applications like CNN based image compression etc.
>
> Regarding interpreting Fig. 4 visually, the ideal quality metric would be on which all the points lie on a straight line. What we do not want is images with multiple levels of distortions giving the same metric score, and vice versa, Image with the same level of distortion giving varying metric scores. If you compare the image in Fig. 4, you can observe that with the high PE result, the vertical spread of points wrt to constant metric scores is reduced, which means that images with the same metric scores are now less variant in perceptual quality, and vice versa. Also, kindly refer to Fig. 5 in the appendix for further results. If you prefer, I can replace Fig. 4 with a plot that is more interpretable.
>
> Regarding Table. 1, I understand the a jargon of number is often difficult to understand, plus its not visually pleasing. However, the experimental techniques and reported evaluation metrics are the standard to analyze perceptual quality metrics. Considering that we analyzed multiple layers of multiple network using three evaluation metrics, graphical representations would have been very difficult. I apologize for the inconvenience. I have added additional explanation in the results section to help readers understand the table better, I hope that helps.
>
> I have added full forms of the terms such as HVS, SISR and added additional explanations for DMOS and SROCC.
>
> Considering firstly that interpretation of learned representations is important and this is the first work to address interpretation of this specific problem (see PROBLEM STATEMENT), secondly , the additional experiment now demonstrating that the analysis can be consequential for practical imaging problems as well, I hope I have sufficiently addressed you major concerns. Thank you again for your time and thorough review, we are at your disposal for any more queries.

---

### Official Review · AnonReviewer4 · 2019-11-04
**Official Blind Review #4**

**Rating:** 3

**Review:**

The submission aims to analyze deep neural network (DNN) features in terms of how well they measure the perceptual severity of image distortions. It proposes to characterize each DNN feature in terms of two well known properties of the human visual system: a) sensitivity to changes in visual frequency and b) orientation selectivity. Both properties are evaluated with respect to the known human Contrast Sensitivity Function (CSF) and measured empirically from the feature’s response to (oriented) sinusoidal gratings. The results are quantified by a composite score termed Perceptual Efficacy (PE).
In a set of comprehensive experiments (several pre-trained DNNs, several layers per DNN and two different datasets of distorted images with human perceptual quality annotations) it is demonstrated that feature representation consisting of a layer’s features with high PE better agree with human perceptual quality judgments than low PE feature representations from the same layer.

I believe the submission convincingly demonstrates a statistical association between the proposed PE score of a DNN feature and human perceptual quality assessments.
Though, it remains unclear whether the characteristics captured by the PE score are necessary or sufficient to explain the success of DNNs to guide image generation tasks by providing a perceptual loss function.
For that I believe it is necessary to demonstrate that the present empirical results can be used to improve results of an image generation task, e.g. super-resolution.
Furthermore the limits of the PE score could be explored by hand-crafting image representations with maximal PE score and comparing their usefulness in guiding e.g. a super-resolution task compared to a pre-trained DNN.

Thus, overall I believe the submission reports interesting initial results but falls short of showing that they capture general properties that can be transferred to improving perceptual loss functions.

**Experience Assessment:**

I have published one or two papers in this area.

**Review Assessment: Checking Correctness Of Derivations And Theory:**

I assessed the sensibility of the derivations and theory.

**Review Assessment: Checking Correctness Of Experiments:**

I assessed the sensibility of the experiments.

**Review Assessment: Thoroughness In Paper Reading:**

I read the paper at least twice and used my best judgement in assessing the paper.

---

> ### Author Response · Authors · 2019-11-06
> **Thank you Reviewer # 4**
>
> Thank you for taking the time to read our manuscript and providing a thorough review. We are pleased that you agree that our proposed analysis and attributes can correlate the ability of pre-trained CNN channels to deliver good perceptual quality features  (as demonstrated by perceptual quality experiments). The prime focus of this paper is to link basic human visual perception to interpret the ability of pre-trained CNN channels as perceptual quality features, a problem that has never been addressed.
>
> Your prime concern is whether the experimental techniques that we have employed (OQA and 2AFC tests) can be considered sufficient evidence to explain the ability of perceptual loss metrics in practical problems, such as super-resolution. An interesting question. We would like to point out that these experimental techniques have been satisfactorily been used to address perceptual loss metrics in prior works such as [1] and are the primary experimental tests when it comes to analyzing the perceptual efficacy of metrics. The loss function provides a singular value that is to be optimized, if the value transitions correspond well with human judgement of perceptual differences, they should be consequential for CNN based Imaging problems. As per [1], superiority in OQA and 2AFC tests should be consequential for practical CNN based imaging problems.
>
> However, to demonstrate, we have included an ADDITIONAL SUPER-RESOLUTION EXPERIMENT in the Appendix (section C). The aim of the experiment is to serve as an indicator that the conclusions of our experimental methods are consequential for; as you have said "image generation tasks such as SR". The additional experimental results should be sufficient to address your major concern, demonstrating that the results are transferable to practical imaging problems. The experiment is not and should not be considered an attempt to surpass state of the art methods, as the primary concern of this paper is understanding and interpretation. Thank you again for your time and thorough review, we are at your disposal for any more queries.
>
> [1]: Zhang, Richard et al. “The Unreasonable Effectiveness of Deep Features as a Perceptual Metric.” 2018 IEEE/CVF Conference on Computer Vision and Pattern Recognition (2018): 586-595.

---

### Decision · Program_Chairs · 2019-12-19

**Decision:**

Reject

**Comment:**

This paper aims to analyze CNN representations in terms of how well they measure the perceptual severity of image distortions.  In particularly, (a) sensitivity to changes in visual frequency and (b) orientation selectivity was used. Although the reviewers agree that this paper presents some interesting initial findings with a promising direction, the majority of the reviewers (three out of four) find that the paper is incomplete, raising concerns in terms of experimental settings and results. Multiple reviewers explicitly asked for additional experiments to confirm whether the presented empirical results can be used to improve results of an image generation. Responding to the reviews, the authors added a super-resolution experiment in the appendix, which the reviewers believe is the right direction but is still preliminary.

Overall, we believe the paper reports interesting findings but it will require a series of additional work to make it ready for the publication.